# Knowledge and Awareness of Risk Factors for HIV Infection and about HIV Testing among Medical Students in Warsaw

**DOI:** 10.3390/v16091470

**Published:** 2024-09-15

**Authors:** Justyna Kowalska, Martyna Cholewik, Carlo Bieńkowski, Aleksandra Maciejczyk, Dominik Bursa, Agata Skrzat-Klapaczyńska

**Affiliations:** 1Department of Adults’ Infectious Diseases, Medical University of Warsaw, 02-091 Warszawa, Poland; justyna.kowalska@wum.edu.pl (J.K.); carlo.bienkowski@wum.edu.pl (C.B.); dominik.bursa@wum.edu.pl (D.B.); agata.skrzat-klapaczynska@wum.edu.pl (A.S.-K.); 2Faculty of Medicine, Medical University of Warsaw, 02-091 Warszawa, Poland; 3IFMSA-Poland Warsaw LC, 02-091 Warszawa, Poland; s077806@student.wum.edu.pl

**Keywords:** HIV, self-testing, students, risk behaviors, late diagnosis

## Abstract

Background: The number of late diagnoses of HIV remains very high in Poland, leading to a high proportion of patients developing and dying of HIV-related diseases. The main reason for this is the very low utilization of HIV testing. Our analyses aimed to investigate knowledge about the indications for HIV testing among medical university students, as well as identify their own HIV testing experiences. Material and methods: A cross-sectional survey study was designed to collect information on the students’ demographics and their experiences of HIV testing, as well as their knowledge of virus transmission and the indications for testing. Data were collected as part of the HIV_week@WUM project conducted at the Medical University of Warsaw in parallel with the 18th European AIDS Conference, which took place in Warsaw between 18 and 21 October 2023. Results: In total, 545 questionnaires were collected. The median age of the study participants was 20 (interquartile range (IQR): 19–22 years). The majority of respondents were as follows: women (67.5%), born in Poland (97.8%), and were attending the medical faculty (56.7%). Only 114 (21.43%) participants had ever been tested for HIV. For all modes of HIV transmission, most of the respondents overestimated the risk of acquiring HIV, but, at the same time, they had never been tested for HIV. Conclusions: Only one in five health sciences students has ever had a test for HIV, which is less than estimates for the general population of Warsaw. There is an ongoing need to popularize testing among future health care providers in order to address both the indications for testing for individuals and the better use of HIV testing in clinical practice.

## 1. Introduction

The number of late HIV diagnoses remains very high in Poland, leading to a high proportion of patients developing and dying of HIV-related diseases such as non-Hodgkin’s lymphoma, cytomegalovirus disease, and *Pneumocystis carinii* pneumonia. As presented in the latest report (2022) on HIV in Poland, performed by the National Health Fund, 20% of deaths due to HIV occur within the first six months of treatment, which can be related to very late diagnoses, when immune deficiency is too severe to save the patient [1]. The issue of late diagnoses poses a serious risk at the population level, as people who have not been diagnosed remain unaware of their infection and, therefore, may continue risky behaviors leading to the onward transmission of HIV to others in the population [2]. The main reason for this is the very low utilization of HIV testing, which can be related to the fear of HIV status disclosure, self-stigmatization, and poor access to testing approaches—especially modern forms—such as self-testing [3]. As a result, even populations that have strict indications for HIV testing are not screened in medical settings [4,5,6]. At the same time, little is known about experiences with HIV testing in the general Polish population, but according to recent analyses by Niedźwiedzka-Stadnik et al., approximately 400,000 HIV tests are taken annually in the general population, which translates into 1% of the population being tested for HIV [7]. In addition, between 2021 and 2022, the number of new HIV cases in Poland more than doubled, including a more than threefold increase in the Mazowieckie Voivodeship [8,9]. Recently, a novel testing approach was made available in Poland, namely, HIV self-testing, which, as a home-based intervention, is believed to decrease the barriers related to social stigma and the fear of disclosure. Local pilot projects conducted in Poland, which included providing self-tests to the partners of people with a newly diagnosed HIV infection, proved that this mode of testing was well accepted [10]. The tests are available to be procured through online pharmacies or distributed free of charge by several non-governmental organizations [11,12]. Our analyses aimed to investigate knowledge about the indications for HIV testing among students, as well as to identify their own HIV testing experiences and to popularize the self-testing approach.

## 2. Material and Methods

A cross-sectional survey study was designed by a group of academic teachers at the Medical University of Warsaw who were experts in HIV. The questionnaire included 13 single-choice, 3 multiple-choice, and 3 open questions on demographics (gender, age, country of origin, academic year, and course of study), experiences with HIV testing, and knowledge of virus transmission and the indications for testing (Appendix A). The questionnaire was pilot tested.

Data were collected as part of the HIV_week@WUM project. Briefly, this was an event promoting HIV testing and awareness among students at the Medical University of Warsaw that was conducted in parallel with the 18th European AIDS Conference taking place in Warsaw between 18 and 21 October 2023. The event aimed to provide education in the area of HIV risks and about the indications for HIV testing. Students were given access to a questionnaire created using the SurveyMonkey software (v. 4.1.1) via a QR code. Students did not participate in any HIV training or discussions directly prior to completing the questionnaire. There were no posters or leaflets on site to suggest answers to the questions. Each participant was asked to complete the form independently only once. Respondents were not obliged to answer all questions. There was no time limit to access the survey. As part of the project, after completing the questionnaire, HIV self-tests were distributed among students as a “learning by doing” experience. Tests were not performed on site, and the test results were not made available to the project team. Students were encouraged to self-test at home and promote this form of testing with others they knew.

### 2.1. Statistical Analysis

As there was no obligation to answer all questions, the proportions of selected responses were calculated according to the number of replies to a specific question. Data were categorized into two groups according to whether respondents declared ever having had an HIV test or not. Baseline characteristics and students’ knowledge were further analyzed by comparing these groups. In terms of year of education, students were categorized into those who had started clinical classes and those who had not. The Mann–Whitney *U* test was used to compare continuous variables, and the chi-squared test was performed to evaluate categorical variables. A *p*-value of <0.05 was considered significant. Statistical analysis was conducted using Quick Statistics Calculators (available at https://www.socscistatistics.com, accessed on 15 February 2024).

### 2.2. Ethical Statement

The project was conducted following the Declaration of Helsinki and approved by the Rector of the Medical University of Warsaw. According to Polish regulations, approval for anonymous survey studies is not required. By completing the questionnaire, each person agreed to anonymously participate in the study and gave consent for publication.

## 3. Results

In total, 545 questionnaires were collected. The median age of the study participants was 20 (interquartile range (IQR): 19–22 years). The majority of respondents were as follows: women (67.5%), born in Poland (97.8%), and attending the medical faculty (56.7%). Most students had not yet decided on their final area of specialty, while others identified surgery (9.54%), anesthesiology (5.87%), and pediatrics (5.87%) as their future specializations. Only 114 (21.43%) participants had ever been tested for HIV. A few respondents knew a person living with HIV (11.44%), but more had suggested that someone should have an HIV test (33.58%). More than half of the participants had heard of HIV self-testing (56.67%), but only 17.07% had heard of the U = U (undetectable = untransmittable) principle (See Table 1).

In terms of knowledge about the risk factors for HIV, 44.30% of the respondents correctly indicated the risk of viral transmission during unprotected anal contact, 29.78% the risk of mother-to-child transmission, and 14.45% the risk of transmission during unprotected vaginal contact. Most participants overestimated the risks. The correct infectious body fluids were selected by 59.14% of participants, those diseases that do not require HIV testing were correctly selected by 30.99%, and the allowable activities of a person on effective retroviral therapy were correctly selected by 25.47%. As many as 92.86% correctly indicated the group of people who should be tested; 51.41% correctly indicated that a person who had performed a risky behavior within the last week should order an HIV test and repeat the test after six weeks in the case of a negative result; and 40.11% correctly indicated that the HIV test should be repeated again after 12 weeks if the second test’s results were negative (see Table 2).

People who had never been tested for HIV were significantly younger than those who had been tested (20, [IQR: 19–22 years] vs. 21, [IQR: 20–23 years]; *p* = 0.0001), were more likely to be female (295/418, 70.57% vs. 66/114, 57.89%; *p* = 0.010), and were students who had not yet begun clinical classes at university (282/417, 67.63% vs. 57/111, 51.35%; *p* = 0.001). They were also significantly less likely to know someone living with HIV (31/418, 7.42% vs. 30/114, 26.32%; *p* = 0.0001), to suggest someone take an HIV test (105/418, 25.12% vs. 73/114, 64.04%; *p* = 0.0001), or to have heard about the U=U principle (60/418, 14.35% vs. 31/113, 27.43%; *p* = 0.001). There were only two significant differences between the two groups regarding knowledge of HIV risk and testing guidelines. People who had never been tested for HIV were less likely to correctly identify the infectious body fluids that enable HIV transmission (236/418, 56.46% vs. 79/114, 69.30%; *p* = 0.013) or the activities in which a person with HIV on effective antiretroviral therapy could engage (99/414, 23.91% vs. 38/114, 33.33%; *p* = 0.042). A comparison of the responses to the multiple-choice questions is shown in Figure 1 and Figure 2.

## 4. Discussion

In our analyses, we identified, from among those students of the Medical University of Warsaw who participated in the survey, that only 20% had ever been tested for HIV. To compare to a broader perspective, in a recent survey performed among young Warsaw citizens, approximately 30% reported that they had been tested for HIV [13]. The lower rate among medical students may be due to low sexual activity or a low number of risky sexual contacts. According to a study by I. Stokłosa et al., medical and paramedical students had the lowest rate of sexual encounters without a condom compared to other academic majors [14]. For the general Polish population, the most current estimates suggest that testing is as low as 1% of the total population [7]. The most commonly provided reason for this is peoples’ self-perception that they are at low or non-existing risk [15]. In a study of Serbian students, 70.9% had positive attitudes towards HIV testing, but only 5.4% had actually been tested [16].

In most of the surveys, people overestimated the risk of acquiring HIV and the number of modes of HIV transmission, while, at the same time, they did not relate this to their own risk behaviors. According to Khawcharoenporn et al., 94% of interviewees with moderate- or high-risk behaviors described themselves as being at no or low risk of HIV infection [17]. In a study by C. Nkwonta et al., a large number of participants did not know that HIV cannot be transmitted by kissing (42.5%), by coughing or sneezing (29%), or by sharing a glass of water (35.8%). In the same study, 46.2% of participants indicated that all HIV-infected pregnant women would give birth to children with acquired immunodeficiency syndrome (AIDS) [18]. Other studies confirmed that a large number of participants were unaware of opportunities to reduce the risk of transmission during childbirth [16,19]. In a survey testing the knowledge of dental students, 50.9% incorrectly identified saliva as an infectious fluid. In addition, 70% overestimated the occupational risks associated with treating people living with HIV [20]. In a study by Milic et al., many study participants believed that the transmission of HIV can happen by using the same hygiene products or cutlery, as well as through insect bites [16]. This finding was also well reflected in our results. Among the respondents, 20.34% identified saliva as an infectious fluid. Viewing saliva as a transmission vector may be partially understandable given the nature of many HIV tests. Many students responded that a person living with HIV cannot practice a medical profession (20.17%), have unprotected sex (66.04%), or give birth naturally (22.08%). In addition, in our study, 85.18% overestimated the risk associated with vaginal contact, 57.30% with the risk of mother-to-child transmission, and 37.57% with anal contact.

People have many beliefs about HIV testing that are not true. In a study by Nkwonta et al., only 35.7% knew that testing one week after unsafe sexual intercourse would not give a reliable result [18]. Babatunde et al. showed that only 53.5% of Nigerian students agreed with the statement that an HIV self-test should be repeated after three months in the case of a negative result [21]. In our survey, only 40.11% correctly selected the answer concerning the need to repeat a negative test.

Many studies have indicated that positive attitudes towards HIV testing are associated with greater knowledge of the virus [16,22,23]. Furthermore, James et al. showed that people who were tested for HIV know more about the virus [24]. In a study by Asante et al., people who were tested for HIV were better at identifying routes of HIV transmission [25]. However, Santanella et al. showed that dental hygienists’ willingness to introduce rapid HIV testing into their practices and provide training to perform such tests did not correlate with their knowledge. [26]. In our study, people who had been tested for HIV were more aware of the types of activities in which a person with HIV on effective antiretroviral therapy could engage (*p* = 0.042) and had better knowledge about the infectious body fluids that enable HIV transmission (*p* = 0.013). However, this group’s knowledge regarding the other questions in the survey was not significantly higher than those who had never tested for HIV.

The results from studies suggest variable attitudes between women and men towards HIV testing. In a study by Asante et al., men were three times more likely to have had an HIV test [25]. In contrast, according to Wang et al., it was female dentist participants who felt it was more necessary to have oral rapid HIV testing [27]. Similar findings were found by Abiodun et al., where women were more likely to want to have an HIV test [23]. In our study, men were more likely to be tested (*p* = 0.010). This may be related to the higher declared number of sexual contacts without a condom in the male student population in Poland [14]. However, the majority of participants in the current study were women. This reflects the structure of students at the Medical University of Warsaw, the vast majority of whom are women. Depending on the study, the 17–20 or 21–25 age groups are identified as those with the highest chance of being tested or wanting to be tested for HIV [23,25]. In our study, those who had been tested were older than those not tested (*p* = 0.0001). In addition, people who had already started clinical classes were more likely to have been tested (*p* = 0.0001). Asante et al. also showed that factors associated with a higher chance of being tested were being single and knowing the facilities where HIV testing was performed [25]. Other factors associated with a higher frequency of testing were, in men, family structure (monogamy), difficulty in confiding in their fathers, and importance placed on religion, and, in women, a greater number of siblings (more than five) and alcohol consumption [28]. In our study, people who knew someone living with HIV (*p* < 0.0001), suggested someone have an HIV test (*p* < 0.0001), or had heard about the U=U principle (*p* = 0.001) were additionally more likely to be tested.

In order to fight the HIV epidemic, multiple strategies are necessary to accurately target interventions for behavioral patterns resulting in taking risks [29]. Awareness campaigns among adolescents are having a positive effect on increasing their knowledge of HIV and how it is transmitted [30]. Even an hour of targeted-testing teaching among medical students increases their confidence in offering someone an HIV test [31]. These introduced strategies should be repeated regularly; otherwise, the effect may be short-lived, as found in research by Cholewinska et al. In that study, medical staff at several hospitals received training on HIV testing. The number of tests performed in the hospitals increased for two months after training but decreased to pre-training levels after a further two months [32]. Increasing the knowledge of HIV among students who are going to work in the broad healthcare area seems to be necessary. There is a need to regularly remind students—and not only during infectious disease classes—about the need for and availability of HIV testing. Further research should focus on developing an intervention to increase HIV knowledge and awareness among medical students. In addition, it would be interesting to examine the HIV testing knowledge of health care professionals and compare it with that of students.

## 5. Limitations

As this was an anonymous survey, a typical bias for this type of research also relates to our work. The system of completing the survey did not require an answer for every question, so some issues had a lower number of responses, but this gave people who were not comfortable with answering all the questions the opportunity to participate. In addition, participants were informed that they would receive an HIV self-test after completing the questionnaire, so their response to the self-test knowledge question may have been biased.

## 6. Conclusions

Only one in five health sciences students had ever had a test for HIV, which is less than estimates for the general population of Warsaw. Male and older students were more likely to have tested for HIV prior to the study. Those who had been tested answered two of the knowledge questions more accurately. Most of the respondents overestimated the risk of acquiring HIV for all modes of HIV transmission, while, at the same time, they had never been tested for HIV. There is an ongoing need to popularize testing among future health care providers in order to address both the indications for testing for individuals and the better use of HIV testing in clinical practice.

## Figures and Tables

**Figure 1 viruses-16-01470-f001:**
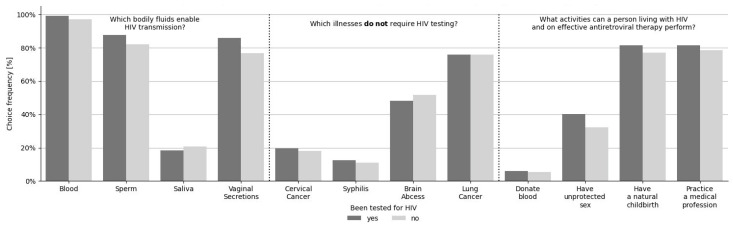
Frequencies of chosen answers for 3 multiple-choice questions depending on HIV testing history.

**Figure 2 viruses-16-01470-f002:**
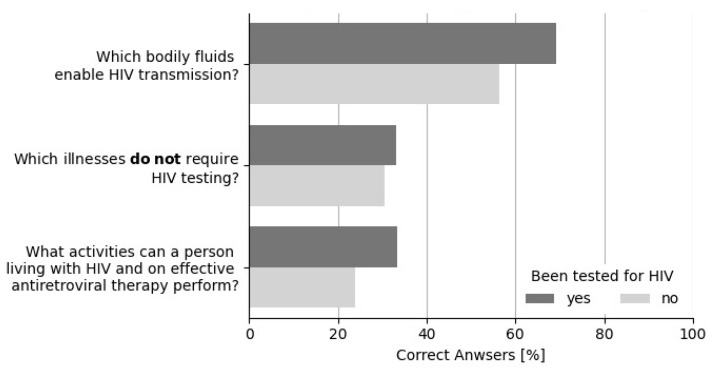
Distribution of correct answers for 3 multiple-choice questions depending on HIV testing history.

**Table 1 viruses-16-01470-t001:** Baseline characteristics of participants stratified by prior testing status.

Characteristic	Total N = 545	Been Tested for HIV N = 114	Never Tested for HIV N = 418	*p*-Value
**Age in years, median [IQR]**	20 [19–22]	21 [20–23]	20 [19–22]	**0.0001**
**Female sex, n (%)**	368 (67.52)	66 (57.89)	295 (70.57)	**0.010**
Polish origin, n (%)	525 (97.77)	113 (100)	400 (97.09)	0.079
Faculty				
Medical, n (%)	309 (56.70)	72 (63.16)	228 (54.55)	0.100
Public health	41 (7.52)	3 (2.63)	38 (9.09)	
Dental medicine	31 (5.69)	5 (4.39)	26 (6.22)	
Medical analytics	29 (5.32)	4 (3.51)	25 (5.98)	
Nursing	23 (4.22)	5 (4.39)	18 (4.31)	
Pharmacy	23 (4.22)	4 (3.51)	18 (4.31)	
Physiotherapy	20 (3.67)	1 (0.88)	19 (4.55)	
Dental hygiene	14 (2.57)	1 (0.88)	13 (3.11)	
Midwifery	14 (2.57)	5 (4.39)	9 (2.15)	
Emergency medicine	12 (2.20)	6 (5.26)	5 (1.20)	
Audiophonology with hearing care	7 (1.28)	1 (0.88)	6 (1.44)	
Electroradiology	6 (1.10)	1 (0.88)	5 (1.20)	
Dietetics	5 (0.92)	2 (1.75)	2 (0.48)	
Dental technology	1 (0.18)	0 (0)	1 (0.24)	
General and clinical logopedics	0 (0)	0 (0)	0 (0)	
Toxicology with elements of forensic science	0 (0)	0 (0)	0 (0)	
**Had not yet begun clinical classes at university, n (%)**	345 (64.01)	57 (51.35)	282 (67.63)	**0.001**
**Knew someone with HIV, n (%)**	61 (11.44)	30 (26.32)	31 (7.42)	**<0.0001**
**Suggested someone take an HIV test, n (%)**	179 (33.58)	73 (64.04)	105 (25.12)	**<0.0001**
Heard of HIV self-testing, n (%)	301 (56.37)	72 (63.16)	228 (54.55)	0.100
**Heard of U = U rule, n (%)**	91 (17.07)	31 (27.43)	60 (14.35)	**0.001**

**Table 2 viruses-16-01470-t002:** Answers to questions about risk factors and guidelines for HIV testing, stratified by prior testing status.

Characteristic	Total N = 545	Been Tested for HIV N = 114	Never Tested for HIV N = 418	*p*-Value
**Multiple-choice questions**				
**Infectious body fluids that enable HIV transmission, n (%)**				
Correct answer: blood, sperm, vaginal secretions	317 (59.14)	79 (69.30)	236 (56.46)	**0.013**
Blood	523 (97.57)	113 (99.12)	406 (97.13)	
Sperm	446 (83.21)	100 (87.72)	343 (82.06)	
Vaginal secretions	422 (78.73)	98 (85.96)	321 (76.79)	
Saliva	109 (20.34)	21 (18.42)	87 (20.81)	
Diseases or conditions not requiring HIV testing, n (%)				
Correct answer: lung cancer	163 (30.99)	37 (33.04)	125 (30.41)	0.059
Lung cancer	400 (76.05)	85 (75.89)	312 (75.91)	
Brain abscess	270 (51.33)	54 (48.21)	213 (51.82)	
Cervical cancer	97 (18.44)	22 (19.64)	74 (18.00)	
Syphilis	59 (11.22)	14 (12.50)	45 (10.95)	
**Activities that a person with HIV on effective antiretroviral therapy can engage in, n (%)**				
Correct answer: have unprotected sex, have a natural childbirth, practice a medical profession	135 (25.47)	38 (33.33)	99 (23.91)	**0.042**
Practice a medical profession	421 (79.43)	93 (81.58)	325 (78.50)	
Have a natural childbirth	413 (77.92)	93 (81.58)	319 (77.05)	
Have unprotected sex	180 (33.96)	46 (40.35)	134 (32.37)	
Be a blood donor	29 (5.47)	7 (6.14)	22 (5.31)	
**Single-choice questions**				
Risk of mother-to-child transmission of HIV if the mother is not treated, n (%)				
Correct answer: 30%	159 (29.78)	42 (36.84)	116 (27.82)	0.062
1%	9 (1.69)	2 (1.75)	7 (1.68)	
10%	60 (11.24)	17 (14.91)	43 (10.31)	
70%	306 (57.30)	53 (46.49)	251 (60.19)	
Risk of HIV transmission through unprotected vaginal contact, n (%)				
Correct answer: 1/10,000 contacts	77 (14.45)	23 (20.18)	54 (12.98)	0.053
0	2 (0.38)	0 (0.00)	2 (0.48)	
1/10 contacts	228 (42.78)	36 (31.58)	190 (45.67)	
1/100 contacts	226 (42.40)	55 (48.25)	170 (40.87)	
Risk of HIV transmission through unprotected anal contact, n (%)				
Correct answer: 3/100 contacts	237 (44.30)	47 (41.23)	190 (45.45)	0.421
0	15 (2.80)	2 (1.75)	13 (3.11)	
3/10 contacts	201 (37.57)	42 (36.84)	158 (37.80)	
3/10,000 contacts	82 (15.33)	23 (20.18)	57 (13.64)	
People who should get tested for HIV, n (%)				
Correct answer: anyone who has performed at least one risky behavior in the past	494 (92.86)	105 (92.11)	387 (93.03)	0.735
Only those performing repeated risky behaviors	23 (4.32)	2 (1.75)	19 (4.57)	
Only people with symptoms of HIV infection	5 (0.94)	4 (3.51)	3 (0.72)	
All children under 5 years of age	10 (1.88)	3 (2.63)	7 (1.68)	
Consultation for a person who performed a risky behavior in the last week, n (%)				
Correct answer: order HIV test; if negative, repeat after 6 weeks	273 (51.41)	52 (45.61)	220 (53.01)	0.162
Order post-exposure prophylaxis	141 (26.55)	35 (30.70)	106 (25.54)	
Too early to order HIV test, postpone the test by 3 weeks	80 (15.07)	24 (21.05)	55 (13.25)	
Order HIV test, end of diagnostics if negative	37 (6.97)	3 (2.63)	34 (8.19)	
Negative HIV self-test result. Should it be repeated? n (%)				
Correct answer: Yes, after 12 weeks	213 (40.11)	53 (46.90)	160 (38.46)	0.105
No	77 (14.50)	17 (15.04)	60 (14.42)	
Yes, after further risky behavior	134 (25.24)	26 (23.01)	107 (25.72)	
Yes, after 48 h	107 (20.15)	17 (15.04)	89 (21.39)	

## Data Availability

The data sets used and/or analyzed during the current study can be made available by the corresponding author upon reasonable request.

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
