# Peer review of "Knowledge and Awareness of Risk Factors for HIV Infection and about HIV Testing among Medical Students in Warsaw"

_viruses, 2024, doi:10.3390/v16091470_

Round 1

Reviewer 1 Report

Comments and Suggestions for Authors

Overview

Thank you for this interesting paper describing the results from a survey exploring HIV testing rates, and knowledge about HIV transmission, in a group of health students based in Warsaw.

It is well written and structured, and I have only minor comments that I hope are helpful.

Introduction

This provides a useful background to the context, highlights the current issues, and provides justification for the study. I do have only one minor quibble, on line 35. People with undiagnosed HIV don’t die of HIV, but an HIV-related disease (due to HIV’s impact on the immune system). A small point, but it’s important to be accurate. I suggest adding a couple of disease examples.

Materials and methods

You provide details about how the participants were recruited, the data collection, and analysis. These seem appropriate for a survey such as this. 

I note that the study received ethical approval. 

Results

There are some interesting findings here, especially given the speciality of the students (health). They give useful insights not only into testing rates, but knowledge of HIV transmission. I note saliva is still identified by some as a transmission vector. In one way this is understandable (they may know some HIV tests are via oral swab), but one would expect health students to know more about the distinction between a virus being present in a particular body fluid, and whether this poses an actual transmission risk. This might be worth comment in the discussion section. 

I also note that relatively few respondents were aware of U=U. 

Discussion/conclusion

You pull your findings together in the discussion, consider the implications, and compare with similar studies elsewhere. You also make recommendations, especially important for a low HIV prevalence country such as Poland. 

Your limitations are appropriate – can you make any suggestions for further research?

One query – in the abstract and in this section, you state (line 238) the respondents “did not relate this to their own risk behaviours”. I couldn’t see in the results that the respondents commented on their own behaviours, but rather they made assessments of risky behaviours in general. Apologies if I missed something, but your wording could be changed if appropriate. 

Reviewer recommendations

1.     In the introduction reword line 35 (and line 14 in the abstract) (“die of HIV”).

2.     Suggest areas for further research. 

3.     Consider rewording line 238 (and line 26 in the abstract) (“their own risk behaviour”).

4.     Consider adding a note in the discussion that viewing saliva as a transmission vector is partly understandable given the nature of many HIV tests.

Author Response

Thank you for the feedback and appreciation of our work. 

Comment 1: In the introduction reword line 35 (and line 14 in the abstract) (“die of HIV”).

Response 1: Thank you for this remark, we applied it to the text. 

Comment 2:  Suggest areas for further research. 

Response 2: Thank you for this remark. We suggested further areas of research at the end of the discussion.  

Comment 3: Consider rewording line 238 (and line 26 in the abstract) (“their own risk behaviour”).

Response 3:  Thank you for this comment. By the phrase “they did not relate this to their own risk behaviors” we meant that they never had an HIV test. In fact, this is inexact, so we have changed the form of the sentence. 

Comment 4: Consider adding a note in the discussion that viewing saliva as a transmission vector is partly understandable given the nature of many HIV tests.

Response 4: Thank you for this remark. We added a sentence about saliva to the discussion.

Reviewer 2 Report

Comments and Suggestions for Authors

Interesting work written a good English. Topic is not so original but this kind of data and results are useful to be present in literature, especially in this period.

Author Response

Thank you for the feedback and appreciation of our work.